# Quality of Life of People with Severe Mental Health Problems: Testing an Interactive Model

**DOI:** 10.3390/ijerph17113866

**Published:** 2020-05-29

**Authors:** Karen Geerts, Ilja Bongers, David Buitenweg, Chijs van Nieuwenhuizen

**Affiliations:** 1GGZ Breburg Institute for Mental Health Care, 5000 AT Tilburg, The Netherlands; k.geerts@ggzbreburg.nl; 2GGzE Institute for Mental Health Care, 5600 AX Eindhoven, The Netherlands; i.l.bongers@tilburguniversity.edu (I.B.); david.buitenweg@ggze.nl (D.B.); 3Scientific Center for Care and Wellbeing (Tranzo), Tilburg University, 5000 LE Tilburg, The Netherlands

**Keywords:** QoL, objective QoL, perceived deficit, people with severe mental health problems

## Abstract

Improvement of subjective quality of life (QoL) is seen as an important treatment outcome in clinical practice. The aim of this study is to test the theoretical model of Cummins, which includes a homeostatic management system. According to this model, objective variables are almost irrelevant to general well-being, while the feeling of having an influence on one’s circumstances (perceived deficit) is related to subjective QoL. The variables of the Cummins model were operationalised based on the Lancashire Quality of Life Profile, a structured interview to assess the subjective QoL of people with severe mental health problems. The Cummins model was tested using structural equation modelling and a mediator model between Objective QoL, Subjective QoL and Perceived Deficit. Subjective QoL and General Well-Being were significantly related and having a meaningful perspective in life was related to General Well-Being. Contrary to the Cummins model, both Objective QoL and Perceived Deficit had a significant relation to Subjective QoL and Perceived Deficit was a partial mediator between Objective QoL and Subjective QoL. Cummins’ theoretical model was partially confirmed. The current study suggests that meaningful (treatment) evaluation of subjective QoL can only be performed if objective QoL, General Well-Being and subjective evaluation (Perceived Deficit and Framework) are taken into account.

## 1. Introduction

Quality of life (QoL) has become a major topic in mental health care. This has to do with a number of fundamental changes in mental health care in the 1990s. First, the focus in mental health has shifted from institutionalizing patients towards community care [1,2,3]. Second, the patient’s own perspective and patient-centred care have become much more important [4,5]. Third, enhancing general well-being is nowadays seen as equally important as the absence of disease symptoms [6,7,8]. Finally, improvement in QoL is seen as an important treatment outcome [9,10,11,12]. Altogether, these changes have resulted in more attention being paid to the influence of psychiatric disorders on daily functioning, on well-being, and on environmental resources [13,14,15]. Although there is no universal definition of QoL, it is generally accepted that it contains objective as well as subjective dimensions [6,14,16,17]. Objective dimensions of QoL relate to objective circumstances such as living situation or finances. Subjective QoL dimensions relate to feelings of well-being and satisfaction. Prior research has focused on unravelling the relationship between objective and subjective QoL, one that seems weak to moderate [6]. Narvaez et al. (2008) for instance, found that psychiatric symptoms were the best independent predictors of subjective and objective QoL in patients with schizophrenia. They also found that the correlation between objective and subjective QoL was low [7]. A review by Priebe et al. (2010) showed higher subjective QoL scores in older patients, those with paid employment and patients with lower symptoms scores. However, the influence of factors—other than age—varied across diagnostic groups. They also found a more consistent association between a higher number of psychiatric symptoms and lower subjective QoL [18].

Hence, numerous objective and subjective factors play a role in the QoL of people with severe mental health problems. Objective measures appear to be more suitable than subjective ones in detecting treatment effects [19,20]. A review of Prince and Prince [17] showed that objective conditions appear to change through political as well as programmatic efforts. Still, improvement of well-being in longitudinal studies of subjective QoL is seldom found. Subjective perceptions appear to be far less tangible than more concrete, objective parameters [21].

The counterintuitive findings on subjective and objective QoL make it difficult to understand the relationship between objective and subjective dimensions. There are two reasons why a better understanding of this relationship is relevant. First, insight into this relationship and other factors that might influence QoL is useful for directing treatment towards enhancing subjective QoL. Second, more insight can help researchers and clinicians to use QoL as an outcome measure of treatment.

Cummins (2000) hypothesised that subjective QoL is governed by a homeostatic mechanism. This means that it normally varies within a quite narrowly defined range, independent of environmental conditions [22]. Cummins argued that when the environmental conditions allow an individual to adapt, there would be little or no relationship between objective and subjective QoL. However, once the threshold for adaptation is exceeded, difficult objective living circumstances begin to drive down subjective QoL [22]. In 2005, Cummins proposed a conceptual model that distinguishes between causal and indicator variables of QoL within the framework of a homeostatic management system. In this model, he described subjective well-being as being the least sensitive because of the homeostatic mechanism [23]. Most people are at least mildly satisfied with their lives. If subjective well-being is already within the normal range, it will be difficult to raise it to higher levels. Cummins (2005) therefore expected the objective variables (objective QoL) to be almost irrelevant to subjective well-being. However, the degree in which someone feels that they can influence these objective circumstances (perceived deficit) does, in the model of Cummins, have an effect on subjective QoL and subjective well-being. If subjective well-being lies below its normative range, there is, for instance, a high probability of depression [23,24].

The aim of this study is to test the Cummins model in a broad population of people with severe mental health problems. To this end, the different variables of the model will be operationalised using the Lancashire Quality of Life Profile. 

## 2. Materials and Methods 

### 2.1. Sample

In this study, eight different data sets of QoL data collected with the Dutch version of the Lancashire Quality of Life Profile (LQoLP) [10] were used. These data sets were collected in previous studies conducted between 1997 and 2014, and included data for 1566 people with mental health problems. The data sets used either the original Dutch version of the LQoLP [25], or the extended Dutch version of the LQoLP [26]. The sample comprised patients with severe mental illness (*n* = 762), forensic psychiatric patients (*n* = 515) and homeless people (*n* = 289). See Table 1 for an overview of the included studies. All procedures performed in studies involving human participants were in accordance with the ethical standards of the institutional and/or national research committee and with the 1964 Helsinki declaration and its later amendments or comparable ethical standards. For this type of study, formal consent is not required.

The LQoLP is a comprehensive instrument to assess the QoL of people with mental health problems. The LQoLP was developed in the United Kingdom [27] and has since been translated into several languages such as German, Italian and Dutch [26]. The interview has frequently been used in scientific research [18,26]. The LQoLP measures an individuals’ satisfaction with 10 different life domains, as well as their general well-being. The LQoLP contains both objective items (‘Do you have a paid job?’) and subjective items (‘How satisfied are you with your monthly income?’). Both the original and extended version of the LQoLP are based on 10 domains: (1) ‘Physical and mental health’, (2) ‘Leisure and social participation’, (3) ‘Finances’, (4) ‘Safety’, (5) ‘Living situation’, (6) ‘Family relations’, (7) ‘Positive Self-Esteem’, (8) ‘Negative Self-Esteem’, (9) ‘Framework’ and (10) ‘Fulfilment’ [25].

Psychometric properties (internal consistency, reliability and validity) of both the original LQoLP and its (extended) Dutch version have been found to be satisfactory [10,25,26]. Van Nieuwenhuizen et al. [26] assessed the internal consistency of the LQoLP and found a Cronbach’s Alpha of 0.93 for the 58-item QoL score. Alpha’s for individual domains ranged between 0.62 and 0.84, with eight out of 10 domains having an Alpha above 0.7. Test-retest reliability of the LQoLP was 0.92. A strong and statistically significant correlation was found between the LQoLP and life satisfaction of 0.73 [26].

### 2.2. Research Variables

Cummins’ model is shown in Appendix A. For the operationalisation of the objective and subjective variables of Cummins’ model, general well-being items, the Negative Affect Scale of the Affect-Balance and the Life Regard Index were used. See Figure 1 for the operationalisation.

#### 2.2.1. Objective Quality of Life

Based on the objective items of six domains of the LQoLP, a latent variable ‘Objective QoL’ was created. The calculation comprised three steps: (1) all objective items were dichotomised; (2) a structural equation model was created; and (3) for each respondent the value of the objective QoL was calculated based on the estimated structural equation model. 

Step 1: The objective items were dichotomised according to the International Classification of Functioning, Disability and Health (ICF) [33]. The ICF is a standardised conceptual system for describing human functioning and problems that can arise from this functioning. Its classification is in line with rehabilitation care [34]. It assumes an integration of the medical and social model. The social model aims for the complete integration of individuals in society. Treatment focuses on the optimal participation on all domains of social life for people with diseases and handicaps. Appendix B shows the items of the LQoLP in relation to the ICF. The objective items of the LQoLP were dichotomised with low level of participation coded 1 and high level of participation coded 2. A high score indicated a better objective QoL.

Step 2: Based on the dichotomised objective QoL items, a latent variable Objective QoL was generated. The latent variable was estimated using a structural equation model, which was composed of all objective QoL items that had a significant relationship with the latent variable Objective QoL.

Step 3: For each individual respondent, the value of the latent variable Objective QoL was calculated using the regression function of the structural equation model. The objective items used and the corresponding score are shown in Appendix C. These items were scored in both the original and extended version of the Dutch LQoLP. For 12 respondents more than three objective items were missing; the Objective QoL score was removed from the data set for these respondents.

#### 2.2.2. Subjective Quality of Life

The Dutch version of the LQoLP consists of 29 subjective items divided over the six domains: physical and mental health, leisure and social participation, finances, safety, living situation and family relations. Each subscale measures patients’ satisfaction on that domain and is rated on a seven-point Likert scale, ranging from 1 (‘life cannot be worse’) to 7 (‘life cannot be better’). In the extended Dutch version of the LQoLP, items were added to the domain ‘Family relations’ (four items) and the domain ‘Safety’ (three items), because of the relatively low reliability of these two domains in the original version. Because of the difference in number of items between the extended and the original version of the LQoLP, the mean score per domain was calculated. The six domains were summed and divided by the number of domains; the mean level of Subjective QoL was used to test the Cummins model.

#### 2.2.3. Perceived Deficit

In this study, Perceived Deficit was calculated using the five LQoLP items that assess a person’s (in)ability to influence their own circumstances. For example, ‘In the past year, have there been times when you wanted to move or improve your living conditions but were unable to do so?’ A mean score was calculated by summing the item scores and dividing the summed score by the number of items. In the analysis, this mean value was used. The feeling of having influence on one’s own circumstances relates to a higher level of QoL and was coded ‘2’, the feeling of not having influence was coded ‘1’. See Appendix D for the description of the incorporated items. Both the original and the extended version of the LQoLP comprised all of these items.

#### 2.2.4. General Well-Being

To measure General Well-Being (GWB), two items of the LQoLP were used. Both at the beginning and at the end of the LQoLP, respondents are asked to rate their life as a whole on theLife Satisfaction Scale, which is a seven-point response scale ranging from ‘1 = can’t be worse’, ‘2 = displeased’, ‘3 = mostly dissatisfied’, ‘4 = mixed (more or less equally satisfied/dissatisfied)’, ‘5 = mostly satisfied’, ‘6 = pleased’, to ‘7 = can’t be better’ [26]. The response scale is identical for both questions. The mean of these items led to a GWB score, which assess how respondents experience their live as a whole [15]. A higher score indicates a higher perceived GWB.

#### 2.2.5. Other Life Issues

In the Cummins model, the most important other life issue mentioned is mood. In the present study, mood was operationalised using the Negative affect scale of the Affect Balance scale [35] and Fulfilment and Framework of the Life Regard Index [36].

The affect questions of the Affect Balance scale can be regarded as equivalent to mood [37]. In the extended version of the LQoLP, respondents were asked whether in the course of the past month they had felt restless, lonely, bored, depressed or upset about being criticised. The five questions of negative affect were measured on a seven-point Likert scale that ranged from ‘cannot be worse’ (1) to ‘cannot be better’ (7). 

The Life Regard Index contains 23 items with a fixed three-point scale (disagree, no opinion, agree). It comprises two sub-scales (Framework and Fulfilment) which together assess the degree to which an individual can envision his life as having some meaningful perspective, or has derived a set of life goals from this perspective [38]. Examples of Framework-items include ‘I feel I have found a really significant basis on which to live my life’, and ‘I have a clear idea what I’d like to do with my life’. Examples of items of the Fulfilment scale are ‘I spend most of my time doing things that are not really important to me’ and ‘I feel I am living a full life’. Higher scores denote having a meaningful life and having goals in life.

### 2.3. Procedure in the Data Files Used

Trained interviewers collected all data. Once the interview has been completed, the interviewer rates the reliability of the interview by selecting one of four Likert options ranging from ‘Very unreliable’ (1) to ‘Very reliable’ (4). The LQoLP does not include a set of grading criteria to assess the reliability, but all interviewers were trained and this training included instructions for assessing the reliability. Interviews that were evaluated as very unreliable were not included in the study. Only data of patients who have given informed consent to use their data for research purposes were included in the database.

### 2.4. Statistical Analysis

All analyses and calculations were conducted using the SPSS package version 19 (IBM Corp, Armonk, NY, United States) and M-plus version 7.2 (Muthén & Muthén, Los Angeles, CA, United States). Descriptive statistics were computed for predictors (Objective QoL, Perceived Deficit, Negative Affect, Framework, and Fulfilment) and for outcomes (Subjective QoL and GWB). Analyses were conducted using the mean values of the variables. Prior to the use of inferential statistics, assumptions of normality, linearity and homoscedasticity were checked and found to be met. A sensitivity analysis was conducted to test whether the final model differed between the original and extended version of the LQoLP. For all analyses, a significance level of *p* < 0.05 was used. 

Simple linear regression analyses were used to test the Cummins model [23]. All models were statistically adjusted for correlations between subjective QoL and GWB and the relationship between Life Regard Index and Affect Balance Scale and GWB. To test the mediation between objective QoL, subjective QoL and perceived deficit, three conditions were tested following the procedures of Baron and Kenny [39,40]. The first step (path c) is to show a significant relationship between the predictor (objective QoL) and the outcome variable (subjective QoL). Next (path a), the predictor and the mediator variable (perceived deficit) must be significantly related. In the third step, the outcome variable is regressed on both the predictor and the mediator. For a mediation effect, the relation between the mediator and the outcome variable (path b) must remain significant, whereas the relation between the predictor and the outcome variable (path c’) must drop to 0 (full mediation), or become less significant (partial mediation). 

## 3. Results

### 3.1. Participants

Participants were predominantly male (72%), with a mean age of 35.16 years (SD = 15.01, range = 12.85). The majority of the sample was unemployed (84.9%), had no intimate relationship (72.2%) and used medication for mental health problems (53.8%). Around one third (27.5%) of the total sample was dissatisfied with their life in general. One-fifth (19.7%) of the group was equally satisfied as dissatisfied. About half (52.8%) of the group was (mostly) satisfied with their life in general. 

The latent variable objective QoL had a mean of 7.94 (SD = 1.63, range = 4.12). Perceived Deficit had a mean of 1.59 (SD = 0.27, range 1.2). The mean of subjective QoL was 4.73 (SD = 0.78, range = 1.7) and the mean of GWB was 4.34 (SD = 1.23, range = 1.7). Table 2 shows all the descriptives, means, standard deviations and ranges of the research variables.

### 3.2. Test of Cummins’ Model 

Mediation between objective QoL, subjective QoL and Perceived Deficit was tested in three paths. Each path was corrected for the correlation between subjective QoL and GWB and the influence of negative affect, fulfilment and Framework on GWB; these are described in the final model.

In the first step, the relation between objective and subjective QoL was investigated (see Table 3). A significant relationship between objective QoL and subjective QoL was identified (path c’: β = 0.277; *p* < 0.001). In step two, the relation between objective QoL (predictor) and Perceived Deficit (mediator) was found to be significant (path b: β = 0.104; *p* < 0.001). In the third step, objective QoL and Perceived Deficit both predicted subjective QoL. The relationship between Perceived Deficit and subjective QoL was significant (path b: β = 0.350; *p* < 0.001), whereas the relationship between objective QoL and subjective QoL was reduced (path c: β = 0.267; *p* < 0.001) compared with the first tested model. Perceived Deficit had an effect on the relation between objective and subjective QoL, thus, Perceived Deficit was a partial mediator; the total effect of objective QoL on subjective QoL was 0.302 (path c’ + (path a * path b)), indicating that two people who differed by one unit in objective QoL were estimated to differ 0.302 units in their reported subjective QoL. 

In the final model, an association between the outcomes subjective QoL and GWB (r = 0.493; *p* < 0.001) was identified. Negative affect (β = 0.035; *p* = 0.112) and fulfilment were both not significantly related with GWB (respectively: β = 0.035; *p* = 0.112; and β = 0.019; *p* = 0.398). Framework had a significant positive effect on GWB (β = 0.208; *p* < 0.001). The results are shown in Table 3. The sensitivity analysis revealed no differences in the interpretation between the final model and the different versions of the LQoLP.

## 4. Discussion

The aim of the present study was to investigate the connection between several factors that influence the QoL of people with severe mental illnesses. To this end, the Cummins (2005) model was used [23]. The results show that in contrast with the Cummins model, both objective QoL and GWB are important predictors of subjective QoL. Perceived Deficit and Framework also have a positive effect on subjective QoL. Hence, Cummins’ theoretical model was partially confirmed.

First, in contrast to Cummins’ model, a significant positive relation was found in the present study between objective QoL and subjective QoL. This finding is in line with a previous study of homeless people: people with the most favourable circumstances reported the highest level of QoL and people who had multiple episodes of homelessness, a criminal record and serious mental health disorders reported lower levels of subjective QoL [41]. As has been noted above, Cummins (2000) argued that when the environmental conditions allow an individual to adapt, there would be little or no relationship between objective and subjective QoL. However, once the threshold for adaptation is exceeded, difficult objective living circumstances of living begin to drive subjective QoL down [22]. The poor circumstances of the patients included in this study (severe mental illness, forensic adolescent psychiatry and homelessness) may have exceeded this threshold for adaptation. 

Second, the relation between Perceived Deficit and subjective QoL was significant in the study. In accordance with the Cummins model, perceived deficit appears to be a partial mediator. Respondents with a higher objective QoL, and who felt to have an influence on their own circumstances, reported a higher subjective QoL. This is in line with the study of Priebe et al. (2010), who found scores that are more favourable in patients with schizophrenia than with other diagnoses [18]. Hayhurst et al. (2014) also showed that QoL rated by people with schizophrenia or by an external assessor differ markedly [42]. Patients with depressive symptoms had better insight and valued their (subjective) QoL less. Perceived deficit could explain these findings; not only objective QoL, but also the perception of these circumstances, influence subjective QoL. Third, Cummins (2005) assumed a positive association between subjective QoL and GWB, and this was indeed found. Neither negative affect nor fulfilment had a significant effect on GWB. This was not in line with previous research, which showed that severity of depression is associated with poor self-reported well-being [43]. Depressive symptoms have been found to be the strongest predictors of poor QoL [44,45]. However, Framework, that is, having some meaningful perspective on life, made a significant contribution to GWB. Pononcy (2016) showed that many persons evaluate their lives in a very positive way, in spite of essential restrictions on their hedonic status [46]. Zika and Chamberlain (1992) also found that psychological well-being and meaning in life are related [47]. They posited that meaning in life could be one of the critical factors in achieving and maintaining a strong sense of GWB. Van Hecke et al. (2017) reviewed theoretical models of QoL and concluded that QoL is a multidimensional construct, composed of objective and subjective dimensions, with an emphasis on the subjective evaluation, which is dynamic in nature and which can be influenced by a variety of factors [14]. The present study supports all of these assumptions.

### 4.1. Clinical Implications

The results of this study show that both objective QoL and perceived deficit have a positive influence on subjective QoL. There is a positive association between Subjective QoL and GWB with a large contribution made by Framework (that is, having a meaningful perspective on life). When improvement of subjective QoL is seen as a treatment outcome, treatment can be focused on improving all of these factors. In practice, logotherapy (based on the premise that the primary motivational force of an individual is to find a meaning in life) [48] or other forms of meaning-focused therapy may be used to help patients obtain meaningful perspectives on their experiences [49]. In addition, treatment may focus on equipping patients with the abilities necessary to overcome the deficits they perceive, for example by offering tips for managing their finances. Alternatively, treatment may focus on helping patients find new goals that are more in line with their opportunities [50]. Overall, the results of this study imply that treatment of people with severe mental health problems should include all facets of life in its attempt to improve the QoL of patients. 

### 4.2. Strengths and Limitations 

Our study has several limitations. First, we targeted modifiable objective dimensions. Therefore, social demographic factors, such as gender and age, were not taken into account. These are potential modifiers of QoL. It would be interesting to test our operationalisation of the Cummins model in several groups to gain insight on the effect of these social demographic factors. Another limitation is that the data are collected over a period of 15 years. Changes in society and in mental healthcare may have influenced the meaning of QoL for people with psychiatric problems, and this might have biased the results. In addition, new circumstances that challenge the mental health of individuals, such as fluctuations in the economic growth, may have occurred. Finally, the analyses were limited to LQoLP data.

A strength of this study is that it contributes to the knowledge of empirical testing of theoretical models in QoL research. Despite the numerous studies on QoL, this is a relatively unexplored area of research and more alternative theoretical models have to be tested. Based on the objective items of the LQoLP’s six domains, a latent variable—Objective QoL—was created. To our knowledge, this has not been operationalised using these different items before. Another positive feature of the study is its use of a large and diverse dataset containing patients with severe mental illnesses. We focused on the most vulnerable group with weak objective circumstances. This might be why the influence of these objective circumstances on subjective QoL was revealed. For research purposes, we propose using the LQoLP as a scoring method for objective QoL. The LQoLP is a comprehensive instrument that is often used in QoL research in psychiatry. The LQoLP describes several factors that influence QoL.

## 5. Conclusions

For the next steps in QoL research, it is important to empirically test various theoretical models. This study shows how this is possible using the LQoLP. Testing several theoretical models in several groups can lead to a better understanding of QoL. In this study, Cummins’ theoretical model was partially confirmed. The study suggests that meaningful (treatment) evaluation of subjective QoL can only be performed if objective QoL, GWB and subjective evaluation (Perceived Deficit and Framework) are taken into account.

## Figures and Tables

**Figure 1 ijerph-17-03866-f001:**
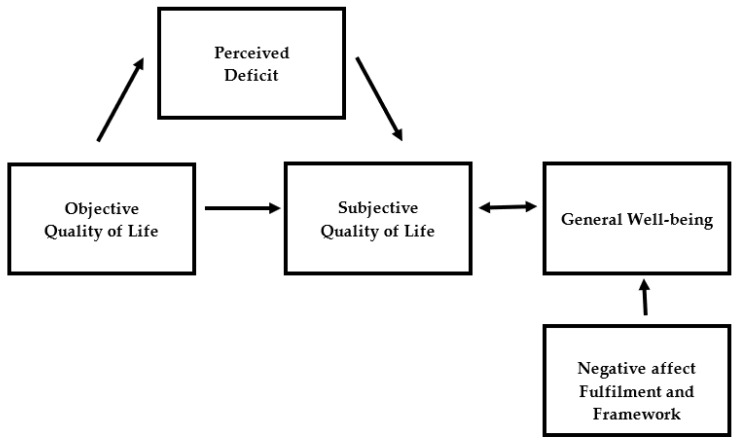
Operationalisation of Cummins’ model using variables of the Lancashire Quality of Life Profile (LQoLP).

**Table 1 ijerph-17-03866-t001:** Study characteristics of eight included studies.

Study	Sample Size	Research Design	LQoLP Version	Research Population
Proost (2002) [27]	116	Cross-sectional	Original	Severe mental illness
Van Nieuwenhuizen et al. (2001) [26]	487	Cross-sectional	Original	Severe mental illness
Barendregt et al. (2015) [28]	172	Longitudinal	Extended	Forensic adolescent psychiatry
De Maeyer et al. (2013) [29]	159	Cross-sectional	Extended	Severe mental illness
Grund et al. (unpublished data)	289	Longitudinal	Extended	Homeless individuals
Bouwman et al. (2008) [30]	135	Cross-sectional	Extended	Forensic psychiatry
Harder et al. (2014) [31]	164	Longitudinal	Extended	Forensic adolescent psychiatry
Van Nieuwenhuizen and Nijman (2009) [32]	44	Cross-sectional	Extended	Forensic psychiatry

LQoLP: Lancashire Quality of Life Profile.

**Table 2 ijerph-17-03866-t002:** Research variables used to operationalise Cummins’ model.

Variable	Mean	SD	Range
Objective QoL	7.94	1.63	4−12
Perceived deficit	1.59	0.27	1−2
Subjective QoL	4.73	0.78	1−7
General Well-Being	4.34	1.23	1−7
Negative affect	3.31	1.75	1−7
Framework			1−3
Fulfilment			1−3

QoL: Quality of life; SD: standard deviation.

**Table 3 ijerph-17-03866-t003:** Linear regression analyses predicting subjective Quality of life (QoL) and General Well-Being.

	Path			β	SE	*p*-Value
Step 1.	Path c	Objective QoL	→ Subjective QoL	0.277	0.021	0.001
		Negative affect	→ General Well-Being	0.031	0.021	0.146
		Fulfilment	→ General Well-Being	0.017	0.021	0.433
		Framework	→ General Well-Being	0.207	0.021	0.001
		Subjective QoL	↔ General Well-Being	0.542	0.018	0.001
Step 2.	Path a	Objective QoL	→Perceived deficit	0.104	0.026	0.001
		Negative affect	→ General Well-Being	0.032	0.021	0.126
		Fulfilment	→ General Well-Being	0.011	0.021	0.612
		Framework	→ General Well-Being	0.222	0.022	0.001
		Subjective QoL	↔ General Well-Being	0.493	0.021	0.001
Step 3.	Path c’	Objective QoL	→Subjective QoL	0.267	0.020	0.001
	Path b	Perceived deficit	→Subjective QoL	0.350	0.021	0.001
	Path a	Objective QoL	→Perceived deficit	0.100	0.026	0.001
		Negative affect	→ General Well-Being	0.035	0.022	0.112
		Fulfilment	→ General Well-Being	0.019	0.022	0.398
		Framework	→ General Well-Being	0.208	0.022	0.001
		Subjective QoL	↔ General Well-Being	0.493	0.021	0.001

SE: Standard Error.

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
