# Peer review of "Quality of Life of People with Severe Mental Health Problems: Testing an Interactive Model"

_ijerph, 2020, doi:10.3390/ijerph17113866_

Round 1

Reviewer 1 Report

General comments:

Overall, I thought this was an interesting study that made good use of an opportunity to operationalise QoL and to empirically test Cummin’s theoretical model. The finding that Objective QoL (and other factors) is an important predictor of Subjective QoL is important, as it suggests important targets for intervention. My specific suggestions are as follows:

  1. Please amend typo in line 23. 'Partial' should be 'partially'.
  2. Change '90's' to '1990's' (line 30) for clarity.
  3. The authors note that 'When more than three objective items were missing for a person, the score was removed from the data set'. (line 124-5). Can the authors explain why they did not impute the missing data and also be clear about the potential bias that this could have introduced. (e.g. if persons with poorer QoL were more likely to have incomplete data, how might this have affected the results? (E.g. In particular, it is probable that individuals with less favourable financial circumstances are more likely to have missing data for these items, so data cannot be assumed to be missing completely at random)
  4. The study uses combined data from the original and extended versions of the LQoLP. The authors note the low reliability of two of the subjective domains of the original version (lines 130-33). I suggest that it may be useful to do a sensitivity analysis to see if the results are consistent if analysis is restricted to the extended (more reliable) version of the LQoLP only.
  5. I would like to know if the authors considered other causal structures for Figure 1. For example, are there circumstances where perceived deficit would confound, rather than mediate, the relationship between Objective and Subjective QoL? For instance, in a situation where the perceived deficit is "In the past year have there been times when you wanted to move or improve your living conditions but were unable to do so?" it is conceivable that this would be a common cause of Objective and Subjective QoL, and thus would be a confounder rather than a mediator. In this situation, the Baron and Kenny test (or any statistical test) would not be informative as a reduction or elimination of the relationship between objective and subjective QoL could indicate either mediation or confounding. Could the authors explain why the causal model proposed is the most plausible, or if either a confounding or mediation model is plausible acknowledge this possibility and it’s implications in the discussion (e.g. in the limitations section).
  6. line 183-4 the authors note: "All models were corrected for correlations between subjective QoL and GWB and the relationship between life regard 184 and affect balance and GWB." Can the authors explain what is meant by 'corrected for' ? Do they mean adjusted for statistically?
  7. Line 217-9 the authors state “the relationship between objective QoL and subjective QoL was significantly reduced (path c: β = 0.267; p = 0.000) compared with the first tested model” [where path c’: β = 0.277]. I question whether this can be classed as a ‘significant’ reduction and suggest that the reviewers should remove the word ‘significant’ from this statement.
  8. I would like to see section 4.1. Clinical Implications expanded. The authors state that treatment should be focused on improving ‘all of these factors’ (i.e. objective QoL, GWB and perceived deficit), but there is no detail on how this might be achieved. I would like to see brief examples of how one might attempt to improve each of these aspects in practice. Improvement of objective QoL may be particularly important among individuals with severe mental illness, since they are often the most disadvantaged in society and may have health conditions that make it more difficult to adapt.
  9. Typo line 293. Change ‘empirical’ to ‘empirically’

Reviewer 2 Report

Thank you for your attempt to test a model suitable for people with serious mental health problems, which is lacking in the area. Please see below a few suggestions to clarify a few things for the readers.

  1. Abstract: It might be good to mention structural equation modelling instead of linear regressions (line 18).
  2. Methods:
    1. Please provide some context of LQoLP for international readers (line 80 onwards). Is it a 'gold standard'? 
    2. Please include the scores of internal consistency, reliability and validity of both versions (line 95) as well instead of just the citations.
    3. Please clarify why a score was removed from the dataset (line 125) as these were highly-stressed people anyway. Removing them may lower the accuracy and sensitivity of the scale?
    4. Please clarify why median score was not used (line 134).
    5. Please clarify how the interviewers estimated the reliability (line 172). Did they use a certain benchmark or grading criteria? Please also comment on their intra- and inter-rater reliability.
  3. Results: Please revise ALL "p=0.000" to "p<0.001" throughout the manuscript because p-value is never zero. Please also include SE in Table 3 (95% CI would be preferable too).
  4. Discussion:
    1. First paragraph of the discussion would be better placed at the last paragraph of the results as it summarises the results in layman terms for the readers in relation to the research aim. 
    2. Whilst noting that gender and age were not taken into account, could the authors not run sensitivity analyses with (and without) these variables and report the finding? Afterall it is important for any validation/testing.
    3. Could also mention that the lack of data on ongoing challenging circumstances on mental health over 15 years as another limitation.
    4. It might be good to mention line 290 in the Introduction section to provide some context.
